# Investigation of Molecular Interactions Mechanism of Pembrolizumab and PD-1

**DOI:** 10.3390/ijms241310684

**Published:** 2023-06-26

**Authors:** Simiao Wang, Faez Iqbal Khan

**Affiliations:** Department of Biological Sciences, School of Science, Xi’an Jiaotong-Liverpool University, Suzhou 215123, China; simiao.wang19@student.xjtlu.edu.cn

**Keywords:** PD-1, pembrolizumab, T cells, cancer, MD simulation

## Abstract

Human programmed cell death protein 1 (PD-1) is a checkpoint protein involved in the regulation of immune response. Antibodies are widely used as inhibitors that block the immune checkpoint, preventing strong immune responses. Pembrolizumab is an FDA-approved IgG4 antibody with PD-1 inhibitory ability for the treatment of melanoma. In this study, we investigated the effect of Pembrolizumab on the conformational changes in PD-1 using extensive molecular modeling and simulation approaches. Our study revealed that during the 200 ns simulation, the average values of the solvent accessible surface area, the radius of gyration, and internal hydrogen bonds of PD-1 were 64.46 nm^2^, 1.38 nm and 78, respectively, while these values of PD-1 in the PD-1/Pembrolizumab complex were 67.29 nm^2^, 1.39 nm and 76, respectively. The RMSD value of PD-1 gradually increased until 80 ns and maintained its stable conformation at 0.32 nm after 80 ns, while this value of PD-1 in the PD-1/Pembrolizumab complex maintained an increasing trend during 200 ns. The interaction between PD-1 and Pembrolizumab led to a flexible but stable structure of PD-1. PD-1 rotated around the rotation axis of the C’D loop and gradually approached Pembrolizumab. The number of hydrogen bonds involved in the interactions on the C and C’ strands increased from 4 at 100 ns to 7 at 200 ns. The strong affinity of Pembrolizumab for the C’D and FG loops of PD-1 disrupted the interactions between PD-1 and PD-L1. Inhibition of the interaction between PD-1 and PD-L1 increased the T cell activity, and is effective in controlling and curing cancer. Further experimental work can be performed to support this finding.

## 1. Introduction

Cancer treatment impacts people’s lives and health [1]. Conventional approaches to cancer treatment usually involve a combination of surgery and chemotherapy. However, the disadvantage of this approach is that the cancer cannot be completely cured and chemotherapy is often associated with severe side effects and pain [2]. Immunotherapy is a treatment that uses the immune system to fight cancer, not by targeting the tumor itself, but by activating or suppressing the patient’s immune system to eliminate cancer cells [3]. The development of immunotherapy has revolutionized cancer treatment and helped improve patient survival rates [4].

Regulation of T cell and B-cell activity by the activation or inhibition of immune checkpoints is a widely used approach in immunotherapy [5]. Antigen-specific and antigen-nonspecific signals play a critical role in regulating immune cell activity. Antigen-specific signaling refers to the recognition of cancer cells by T cells through the binding of the T cell receptor (TCR), which is expressed on the surface of T cells, to the major histocompatibility complex (MHC) on the surface of cancer cells [6]. In contrast, antigen-nonspecific signaling refers to the synergistic action of the CD28 protein expressed on the surface of T cells with the TCR to regulate T cell activity [7].

Monoclonal antibody therapies targeting the interaction between Programmed cell death protein 1 (PD-1) and its ligands, Programmed death- ligand 1 (PD-L1) and Programmed death- ligand 2 (PD-L2), have shown remarkable efficacy in treating and curing cancer [8,9]. PD-1 belongs to the CD28 family and is commonly found on the cell membrane surface of activated T cells and B cells [10]. It recognizes endogenous PD-L1 or PD-L2, activating the corresponding signaling pathways to suppress the immune response of T cells [11]. However, PD-L1 and PD-L2 are also commonly present on the surface of some cancer cells, giving them some ability to mount an immune response [12]. PD-1 inhibitors have a high affinity for PD-1 and can compete with endogenous PD-L1 (Figure 1). This leads to an increase in the activity of T cells in destroying cancer cells, which in turn leads to changes in the overall immune system of the patient, possibly causing some autoimmune diseases [13]. Studies of the crystal structure of mouse and human PD-1/PD-L1 have revealed that the PD-1/PD-L1 interaction is based on the CC’ loop and the FG loop of PD-1 [14].

Pembrolizumab, an IgG4 antibody with PD-1 inhibitory activity, was approved by the FDA in 2014 for the treatment of melanoma [15]. The study of the crystal structure of the PD-1/Pembrolizumab complex provided a basis for the exploration of protein–protein interaction (PPI) [16]. However, the knowledge gained from this analysis is limited. The use of molecular dynamics (MD) simulations provides a unique perspective that can be used to further refine our investigations. Conventional crystallization methods provide only a static snapshot of the PPI, whereas molecular dynamics simulations represent a dynamic process that can reflect possible conformational changes in the protein [17,18]. In addition, free energy analyzes based on molecular dynamics simulations can help identify key residues involved in PPI [19]. These data are important for the development of small molecules or peptide inhibitors targeting PD-1. In this article, we present a comprehensive analysis of the interaction mechanism between PD-1 and Pembrolizumab. To this end, molecular dynamics simulations of the PD-1/Pembrolizumab complex were performed using the GROMACS program for a duration of 200 ns. A number of computational methods were used to determine the plausible binding mechanism between PD-1 and Pembrolizumab. Our goal is to contribute to future research on PD-1 and the interaction methods of the antibody Pembrolizumab.

## 2. Results

### 2.1. Structure Analysis of PD-1 Complex

The complex structure of PD-1 and Pembrolizumab was obtained from the Protein Data Bank (PDB: 5B8C) and missing atoms were added to the structure. Pembrolizumab consists of light and heavy chains. PD-1 is a β-sandwich structure with nine loops, of which the FG loop and the C’D loop play the major part in the interaction of PD-1 with Pembrolizumab. The PD-1 has 2 β-sheets, 3 β-hairpins, 2 β-bulges, 10 strands, 1 α-helix, 14 β-turns, 1 γ-turn, and 1 disulphide bond. The light chain of Pembrolizumab has 3 β-sheets, 5 β-hairpins, 3 β-bulges, 12 strands, 1 α-helix, 15 β-turns, 1 γ-turn, and 1 disulphide bond, whereas the heavy chain of Pembrolizumab has 2 β-sheets, 4 β-hairpins, 4 β-bulges, 11 strands, 3 α-helices, 10 β-turns, 1 γ-turn, and 1 disulphide bond. The total secondary structural elements present in the light and heavy chains of Pembrolizumab and PD-1 are shown in Figure 2. It was found that the PD-1 forms hydrogen bonds such as Ser_60_-Tyr_34_, Glu_61_-Tyr_57_ and Leu_128_-Tyr_53_, and van der Waals interactions such as Glu_61_-Gly_33_ and Glu_61_-Tyr_34_ with the light chain of Pembrolizumab. A salt-bridge Arg_86_-Arg_96_ was present between PD-1 and light chain of Pembrolizumab. Several strong hydrogen bonds, van der Waals interactions, and a salt bridge existed between PD-1 and heavy chain of Pembrolizumab (Table 1).

### 2.2. Structure Conformation of PD-1/Pembrolizumab Complex

To investigate the structural conformations, the simulated structures of the PD-1 and PD-1/Pembrolizumab complexes were analyzed at 100 ns, 150 ns, and 200 ns (Figure 3). It can be seen that the BC loop, the C’D loop, and the FG loop of PD-1 in the PD-1/Pembrolizumab complex have almost no conformational differences at the three time scales, while some other loops have large conformational differences. All loops in PD-1 show strong fluctuations and divergences, while these loops in the PD-1/Pembrolizumab complex are stable and show the least divergence. The C’D loop of PD-1 in the PD-1/Pembrolizumab complex was found to be relatively distant from the β-sheet and in contact with the Pembrolizumab heavy chain. The FG loop of PD-1 in the PD-1/Pembrolizumab complex shows almost no changes, whereas some deviations in the FG loop have been reported in the case of PD-1. The AB loop in PD-1 and the PD-1/Pembrolizumab complex are not stable. The results suggest that the binding of PD-1 with the antibody Pembrolizumab leads to stabilization of the C’D and FG loops. This strong affinity of Pembrolizumab for the C’D and FG loops of PD-1 leads to the disruption of the PD-1 and PD-L1 interactions. In addition, the FG and A’B loops experience a shift of ~6.5 Å and ~15.5 Å, respectively, during the 200 ns MD simulations.

### 2.3. Interaction of Pembrolizumab and PD-1

To investigate the interaction mechanism of PD-1 with Pembrolizumab, the simulated structures of the PD-1/Pembrolizumab complex were analyzed at 100 ns, 150 ns, and 200 ns. Interactions occur mainly at three different sites on PD-1, namely in the FG loop, in the C’D loop and near the CC’ loop. Residues Ser_60_, Glu_61_ and Phe_63_ in the BC loop are only involved in the interaction between PD-1 and the light chain of Pembrolizumab with Tyr_34_ and Tyr_57_ in the initial state (Figure 4a). Residues Leu_128_, Lys_131_, and Ala_132_ of the FG loop in PD-1 form hydrogen bonds with residues Tyr_53_, Tyr_57_, and Leu_58_, Val_62_ of the Pembrolizumab light chain during different time scales (Figure 4b–e).

The C’D loop contributes to a large number of hydrogen bonds in the interaction of PD-1 with Pembrolizumab. The C’D loop residues such as Arg_86_, Ser_87_, Gln_88_, Pro_89_, Gly_90_ and Asp_92_ form hydrogen bonds with residues Thr_35_, Asn_52_, Asn_55_, Gly_57_, Thr_58_, Asn_59_, Arg_99_, Phe_103_, and Asp_104_ at 200 ns of Pembrolizumab (Figure 5a–d). CC’ loop regions are also involved in the interaction with the heavy chain of Pembrolizumab. The C strand and C’ strand regions allow more interactions with PD-1 Asn_66_, Tyr_68_, Arg_69_, Asn_74_, Gln_75_, Thr_76_, Asp_77_, and Lys_78_ due to the presence of side chain Thr_30_, Asn_31_, Tyr_33_, Ser_54_, Asn_55_, Tyr_101_, Arg_102_, Phe_103_, which may contribute to the stability of PD-1 (Figure 5e–h).

### 2.4. Structure Deviations and Compactness

The root mean square deviation (RMSD) is one of the most important properties to determine whether a protein is stable and close to its experimental structure. To better determine the interaction mechanism of PD-1 with Pembrolizumab, we first determined the RMSD values of PD-1 and PD-1 in the PD-1/Pembrolizumab complex. The RMSD value of PD-1 fluctuates upward during the first 80 ns and maintains a stable average value. The average RMSD values during the stable region (150–200 ns) of PD-1 and PD-1 in the PD-1/Pembrolizumab complex were 0.32 nm and 0.33 nm, respectively (Figure 6b). It was found that PD-1 has rather random fluctuations and the binding of Pembrolizumab leads to the minimization of the average fluctuations. The root mean square fluctuation (RMSF) is also commonly used to assess flexibility. The results show that there is a peak at Lys_131_ of PD-1, the part of the FG loop, while PD-1 in the PD-1/Pembrolizumab complex remains in a lower steady state. This suggests that binding to Pembrolizumab makes the FG loop more stable.

A protein’s solvent accessible surface area (SASA) refers to the area of its surface that interacts with its solvent molecules. The average SASA of PD-1 and PD-1 in the PD-1/Pembrolizumab complex was 64.46 nm^2^ and 67.29 nm^2^, respectively (Figure 6c). Binding of Pembrolizumab to the loop of PD-1 leads to an increase in the SASA of PD-1. An increase in SASA may lead to the suppression of the effect of PD-1. In addition, the mean radius of gyration (*R_g_*) values of PD-1 and PD-1 in the PD-1/Pembrolizumab complex were found to be 1.38 nm and 1.39 nm, respectively (Figure 6d). The *R_g_* is a parameter linked to the tertiary structural volume of a protein and has been applied to obtain insight into the stability of the protein in a biological system. The *R_g_* values suggest that the binding of Pembrolizumab could slightly relax the structure of PD-1. 

### 2.5. Analysis of Intermolecular and Intramolecular Hydrogen Bonds

Hydrogen bond analysis is a widely used method to study protein–protein interactions. The average number of hydrogen bonds in PD-1 and PD-1 in the PD-1/Pembrolizumab complex was reported to be 78 and 76, respectively. The *R_g_* results indicate that the PD-1 has a less compact structure (Figure 7a). This might be due to the fact that PD-1 has more loops that can move freely. The average number of hydrogen bonds between Pembrolizumab and three specific regions of the PD-1 protein, namely, the FG loop, the CD loop, and the C and C’ strands were found to be 1, 4, and 4, respectively. (Figure 7b). The number of hydrogen bonds on FG loop and C’D loop remained stable after 80 ns time scale. In contrast, the number of hydrogen bonds on the C strand and C’ strand starts to increase at 150 ns, and eventually increases from 4 at 100 ns to 7 at 200 ns. This may indicate that the number of amino acids involved in the C strand and C’ strand to bind to Pembrolizumab increases.

### 2.6. Secondary Structural Analysis

It was found that the average residues involved in the PD-1 total structure, coil, β-sheet, β-bridge, bend, and turn are 57%, 25%, 49%, 1%, 17%, and 7%, respectively. On other hand, the average residues involved in the PD-1/Pembrolizumab total structure, coil, β-sheet, β-bridge, bend, and turn are 61%, 25%, 50%, 0%, 13%, and 11%, respectively (Table 2). A secondary structure analysis of PD-1 and PD-1/Pembrolizumab complex showed that the interaction with Pembrolizumab resulted in a decrease in bends and β-bridges and an increase in the total amount of secondary structure, β-sheet and turn. The increased percentage of amino acids involved in secondary structure formation enhances the stability of the protein structure, leading to inhibition of the interaction between PD-1 and PD-L1. The secondary structure of the light and heavy chains of Pembrolizumab is also shown (Figure 8a–d).

### 2.7. Gibbs Free Energy Landscape

The Gibbs free energy (GFE) landscape provides information about protein folding and states. The Gibbs free energy landscape was computed using *gmx covar*, *gmx anaeig*, and *gmx sham* using the projections of their own first (PC1) and second (PC2) eigenvectors, respectively. The closer the color is to blue, the lower the free energy. Multiple peaks of low free energy indicate that the protein can switch between different conformations. The GFE landscape of Pembrolizumab light chain, Pembrolizumab heavy chain, PD-1 in the PD-1/Pembrolizumab complex and PD-1 were plotted separately (Figure 9a–d). PD-1 has a larger area of lower free energy regions, and PD-1/Pembrolizumab complex have multiple peaks. This suggests that PD-1 may undergo structural changes more readily.

### 2.8. Principal Component Analysis

PCA is a powerful statistical analysis technique that can show trends, jumps, clusters and outliers by reducing the dimensionality of a data set with a high number of dimensions. The 2D projections of trajectories onto eigenvectors at a different time scale are reported. Interaction of PD-1 with Pembrolizumab results in significant changes in the collective motion of PD-1. During the 100–200 ns MD simulations, the eigenvalues of PD-1 and PD-1 in the PD-1/Pembrolizumab complex were found to be 535.66 nm^2^ and 408.96 nm^2^, respectively (Figure 10). The eigenvector trajectory of PD-1 was consistently higher than that of PD-1 in PD-1/Pembrolizumab. The eigenvector 1 and eigenvector 2 of PD-1 and PD-1 in PD-1/Pembrolizumab complex were analyzed and recorded to understand atomic fluctuations and intramolecular collective motions, respectively. The results show that the atomic fluctuations in PD-1 occur in a wider range. In contrast, most atoms of PD-1 in the PD-1/Pembrolizumab complex are relatively stable, with only a few specific loops undergoing large deformations. Most of them are involved in the interaction of PD-1 with Pembrolizumab. This means that the presence of Pembrolizumab makes the structure of PD-1 more stable.

### 2.9. Mechanism of PD-1 and Pembrolizumab Interaction

The 200 ns simulation revealed a displacement of PD-1 and Pembrolizumab. By comparing the structure at 200 ns with the initial state, it was observed that PD-1 rotated as a whole around the axis of the C’D loop in relation to Pembrolizumab (as shown in Figure 11a). Additionally, it was observed that after 200 ns of simulation, the FG loop of PD-1 occupied a binding site on Pembrolizumab, which is the binding site of the BC loop of PD-1 at the starting state (Figure 11b).

## 3. Discussion

The immune checkpoint inhibitor agents could reactivate cytotoxic T cells to work against cancer cells. The FDA approval of PD-1 inhibitors (Pembrolizumab) have diversified the clinical activity toward a wide variety of solid tumors including lung cancer, renal cell cancer, and ovarian cancer [20]. A comprehensive analysis of the interactions of the immune checkpoint blockers can provide a better understanding of their therapeutic mechanisms of action [21]. We investigated the effect of Pembrolizumab on the conformational changes in PD-1 using extensive molecular modeling and simulation approaches [22]. During the 200 ns simulation, the relative position and binding sites of PD-1 changed significantly after Pembrolizumab binding. The structural conformation of PD-1 and PD-1 in the PD-1/Pembrolizumab complex shows the interaction with Pembrolizumab makes the loops of the PD-1 binding domain more stable. By comparing the average value of intramolecular hydrogen bonds, the radius of gyration and SASA of PD-1 and PD-1 in the PD-1/Pembrolizumab complex, it was found that the binding of Pembrolizumab makes PD-1 more flexible. By comparing the structures of PD-1 and PD-1/Pembrolizumab complex, we propose that the alteration in the binding site is a consequence of the rotation of PD-1 along the C’D loop axis. CC’ loop region is an essential binding region of PD-1 for Pembrolizumab. The number of hydrogen bonds between the CC’ loop region of PD-1 and Pembrolizumab gradually increased during the simulation. This might be due to the rotation of PD-1. At the same time, the rotation of PD-1 causes the FG loop to occupy the binding site belonging to the BC loop. This also allows the FG loop to bind more stably to Pembrolizumab.

## 4. Materials and Methods

### 4.1. Molecular Modeling

The crystal structure of the PD-1/Pembrolizumab complex (PDB: 5B8C) was obtained from the Protein Data Bank [23]. The missing atoms in the structure of PD-1 were modeled using Modeller [24]. The complete protocols are available in previous publications [25,26,27,28,29]. Structural analysis was performed using PDBsum [30] and numerous modules of MD simulations. Discovery Studio [31] and Pymol [32] were used to visualize and draw the structure.

### 4.2. Molecular Dynamics Simulations

Molecular dynamics simulations were performed on PD-1 and PD-1/Pembrolizumab complex using GROMACS 2018.2 using a standard protocol [33,34,35,36]. Atomic positions were first constrained using a GROMOS96 43a1 force field file. Energy minimization was achieved using the steepest descent algorithm. Two levels of equilibration were performed: NVT (constant number of particles, volume and temperature) and NPT (constant number of particles, pressure and temperature). The details of the methodology of MD are explained in previous publications [37,38].

#### 4.2.1. Root Mean Square Deviation

Root mean square deviation (RMSD) is a fundamental property extensively employed in the analysis of protein structures. It serves as a valuable metric to quantify the differences between the experimental and simulated protein structures obtained from molecular dynamics simulations. The RMSD value enables the assessment of the accuracy and reliability of the MD simulation results, and thereby aids in the interpretation of the structural dynamics and conformational changes that occur during the simulation.
(1)$$RMSD\left(t\right)={\left[\frac{1}{M}\sum_{i=1}^{N}m_{i}{|r_{i}\left(t\right)-r_{i}^{ref}|}^{2}\right]}^{\frac{1}{2}}$$
where *M* denotes the mass of protein and $m_{i}$ denotes the atomic mass of *i*.$r_{i}\left(t_{j}\right)$ denotes the location of atom *i* at moment *t*. $r_{i}^{ref}$ denotes the location of atom *i* in reference structure.

#### 4.2.2. Root Mean Square Fluctuation

The assessment of the fluctuations of atoms in a protein is crucial in the analysis of MD simulations. The root mean square fluctuation (RMSF) is a widely employed metric for quantifying the displacement of a specific atom or group of atoms in the protein structure, relative to a reference structure, based on MD simulation results.
(2)$${RMSF}_{i}={\left[\frac{1}{T}\sum_{t_{j}=1}^{T}{\left|r_{i}\left(t_{j}\right)-r_{i}^{\mathrm{ref}\;}\right|}^{2}\right]}^{\frac{1}{2}}$$
where *T* denotes the simulation duration. $r_{i}\left(t_{j}\right)$ denotes the location of atom *i* at moment *j*. $r_{i}^{ref}$ denotes the location of atom *i* in reference structure.

#### 4.2.3. Radius of Gyration

The atomic distribution of the target protein can be assessed using the radius of gyration (*R_g_*), which quantifies the distance between each atom in the protein and its center of mass during rotation. In addition, *R_g_* is frequently employed as an indicator of the compactness of a protein.
(3)$$R_{g}={\left(\frac{\Sigma_{i}{\left|r_{i}\right|}^{2}m_{i}}{\Sigma_{i}m_{i}}\right)}^{2}$$
where $r_{i}$ denotes the distance between atom *i* and the center of mass of the target protein, while $m_{i}$ represents its mass.

### 4.3. Principal Component Analysis

Principal component analysis (PCA) is a technique for reducing the dimensionality of large data sets, thereby increasing interpretability while minimizing information loss. The diagonalized covariance matrix is the most appropriate decomposition method.
(4)$$C_{ij}=(r_{i}-\langle r_{i}\rangle )\times (r_{j}-\langle r_{j}\rangle )(i,j=1,2,3,\ldots ,N)$$

*C* represents covariance matrix, *N* represents the total number of C_α_ atoms. *r_i_* represents the Cartesian coordinate of *i*th C_α_ atom. <*r_i_*> represents the average Cartesian coordinate in all time steps.

### 4.4. Gibbs Free Energy Landscape

The GFE landscape shows the conformational changes of the PD-1 and PD-1/Pembrolizumab complex.
(5)$$G_{\left(PC1,PC2\right)}=-k_{B}T\ln P_{\left(PC1,PC2\right)}$$

Principal component 1 (PC1) represents the linear combination of the variables that extract the maximum variance. Principal component 2 (PC2) represents the vertical linear combination with PC1. *k*_B_ represents the Boltzmann constant; *T* represents temperature.

## 5. Conclusions

This research has contributed to a comprehensive understanding of the dynamic binding mechanism between PD-1 and Pembrolizumab. In this study, 200 ns molecular dynamics simulations were carried out to evaluate the dynamic binding between PD-1 and Pembrolizumab. The results showed that PD-1 became more stable and flexible after interaction with Pembrolizumab. In addition, the number of residues involved in the PD-1/Pembrolizumab interaction on the C and C’ strands in PD-1 gradually increased with the simulation time. This resulted in a change in the conformation of the protein complex and the relative interaction of PD-1 and Pembrolizumab. In addition, the number of intermolecular hydrogen bonds between PD-1 and Pembrolizumab increased, resulting in a more stable interaction between PD-1 and Pembrolizumab.

## Figures and Tables

**Figure 1 ijms-24-10684-f001:**
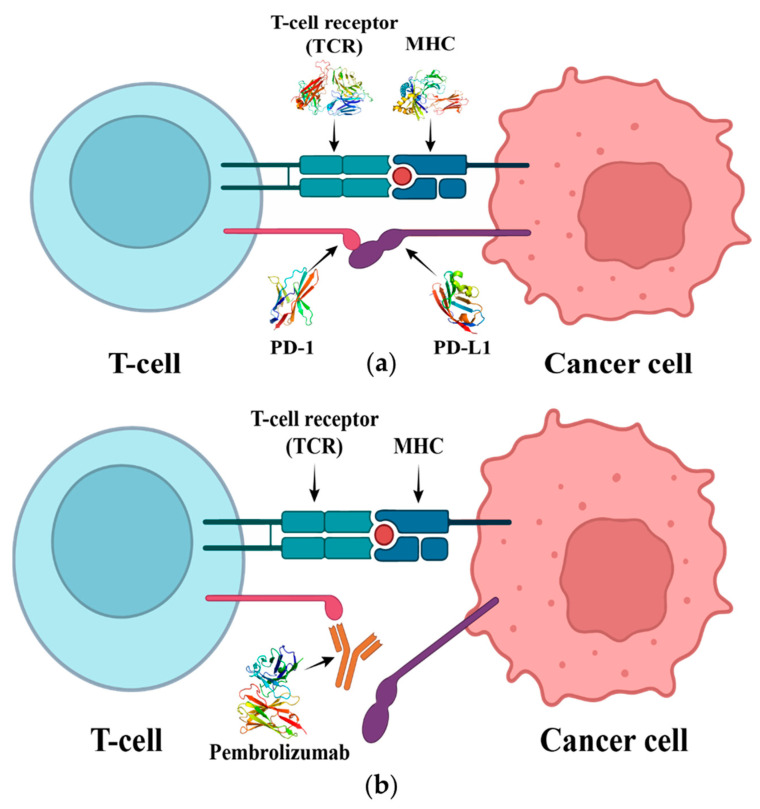
(**a**) The interaction of T cell receptor (TCR) with the major histocompatibility complex (MHC). It initiates T cell activation, and the interaction of PD-1 with PD-L1 will lead to the inhibition of T cell. (**b**) Pembrolizumab blocks the connection between PD-1 and PD-L1, thus preventing T cell inhibition and restoring its activity.

**Figure 2 ijms-24-10684-f002:**
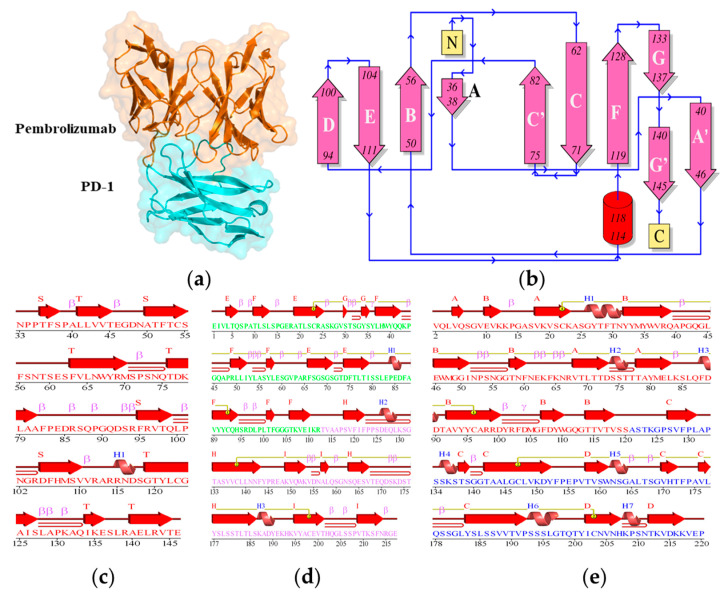
(**a**) The crystal structure, and (**b**) the topology of the PD-1/Pembrolizumab complex. The secondary structure elements of (**c**) the PD-1, (**d**) the light-chain of Pembrolizumab and (**e**) the heavy-chain of Pembrolizumab.

**Figure 3 ijms-24-10684-f003:**
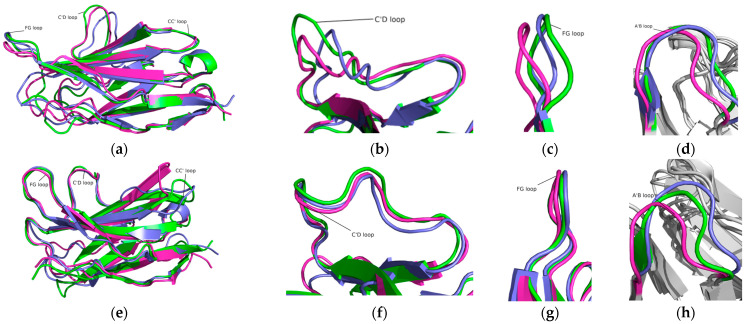
Structural dynamics of the PD-1 and PD-1/Pembrolizumab complex at 100 ns (green), 150 ns (slate), and 200 ns (magenta). (**a**) PD-1, (**b**) C’D loop in PD-1, (**c**) FG loop in PD-1, and (**d**) A’B loop in PD-1. (**e**) PD-1 in the PD-1/Pembrolizumab complex, (**f**) C’D loop in the PD-1/Pembrolizumab complex, (**g**) FG loop in the PD-1/Pembrolizumab complex, and (**h**) A’B loop in the PD-1/Pembrolizumab complex.

**Figure 4 ijms-24-10684-f004:**
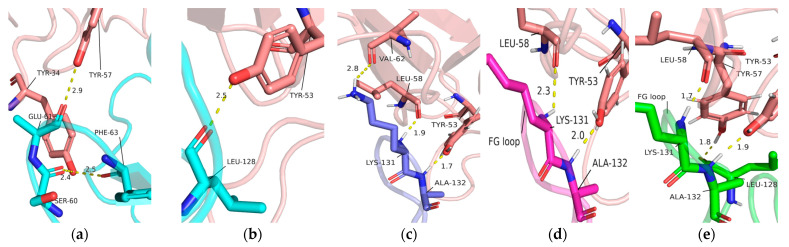
(**a**) Interaction details of light chain of Pembrolizumab (tints) with BC loop at initial state. Interaction details of light chain of Pembrolizumab (tints) with FG loop of PD-1 at (**b**) initial state (cyan), (**c**) 100 ns (green), (**d**) 150 ns (slate), and (**e**) 200 ns (magenta) during MD simulations.

**Figure 5 ijms-24-10684-f005:**
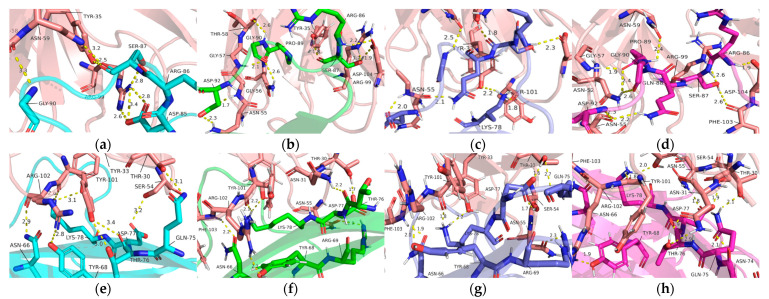
Interaction details of heavy chain of Pembrolizumab (tints) with C’D loop at (**a**) initial state (cyan), (**b**) 100 ns (green), (**c**) 150 ns (slate), and (**d**) 200 ns (magenta). The C strand and C’ strand region of PD-1 at (**e**) initial state (cyan), (**f**) 100 ns (green), (**g**) 150 ns (slate), and (**h**) 200 ns (magenta) during MD simulations.

**Figure 6 ijms-24-10684-f006:**
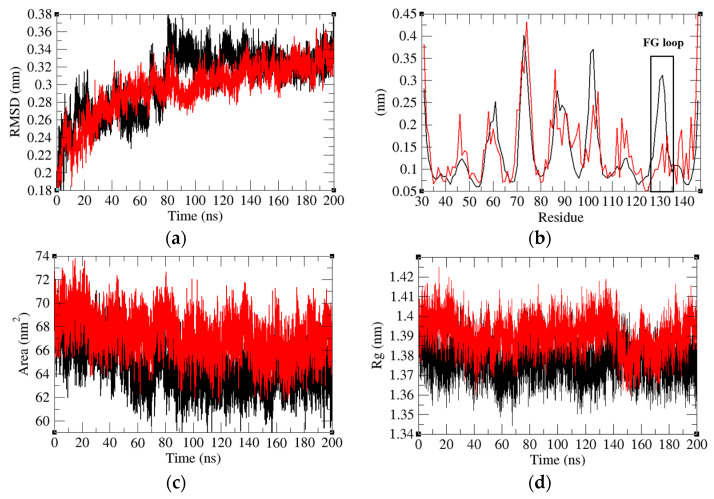
(**a**) Root mean square deviations, (**b**) root mean square fluctuations, (**c**) solvent accessible surface area, and (**d**) radius of Gyration of PD-1 (black) and PD-1 in the PD-1/Pembrolizumab complex (red).

**Figure 7 ijms-24-10684-f007:**
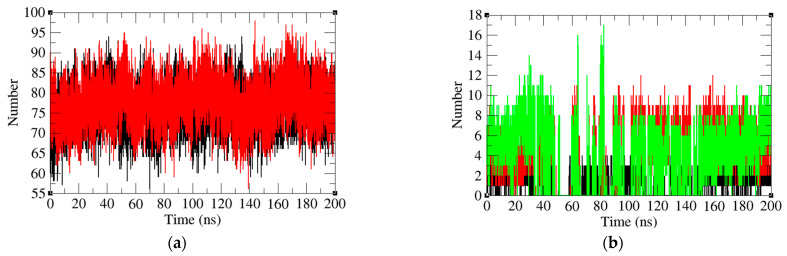
(**a**) Intramolecular hydrogen bond number of PD-1 (black) and PD-1 in PD-1/Pembrolizumab complex (red). (**b**) Hydrogen bond number between Pembrolizumab and the FG loop (black), the C strand and the C’ strand (green) and C’D loop (red).

**Figure 8 ijms-24-10684-f008:**
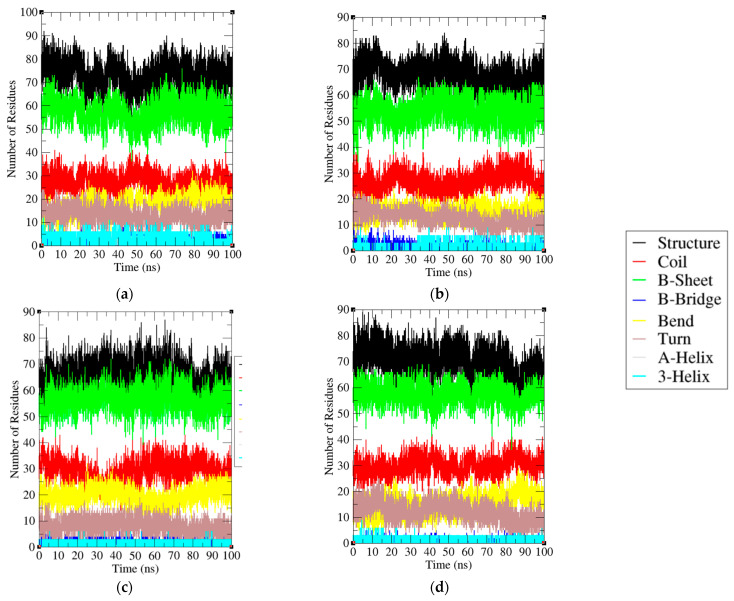
Secondary structure of (**a**) Light chain of Pembrolizumab, (**b**) Heavy chain of Pembrolizumab, (**c**) PD-1, and (**d**) PD-1 in PD-1/Pembrolizumab complex during 200 ns MD simulation. Total structure, coil, β-Sheet, β-Bridge, Bend, Turn, α-Helix and 3-Helix are represented by black, red, green, blue, yellow, brown, silver and cyan, respectively.

**Figure 9 ijms-24-10684-f009:**
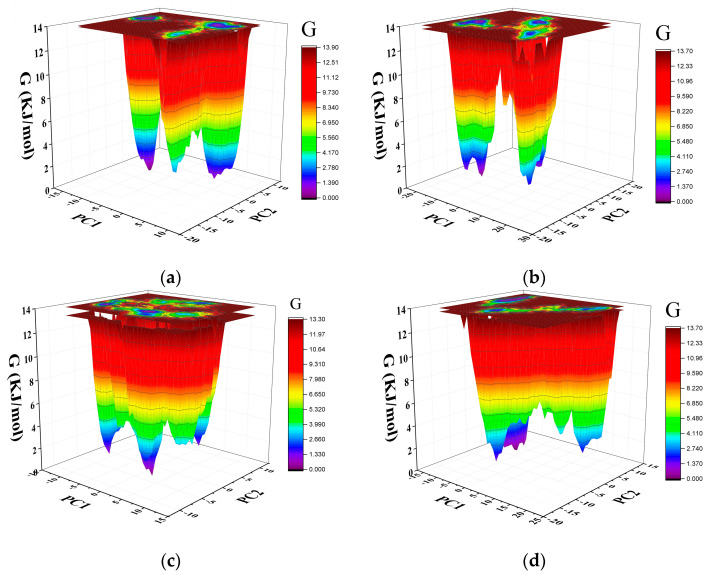
The GFE landscape of (**a**) Pembrolizumab light chain, (**b**) Pembrolizumab heavy chain, (**c**) PD-1, and (**d**) PD-1 in PD-1/Pembrolizumab complex.

**Figure 10 ijms-24-10684-f010:**
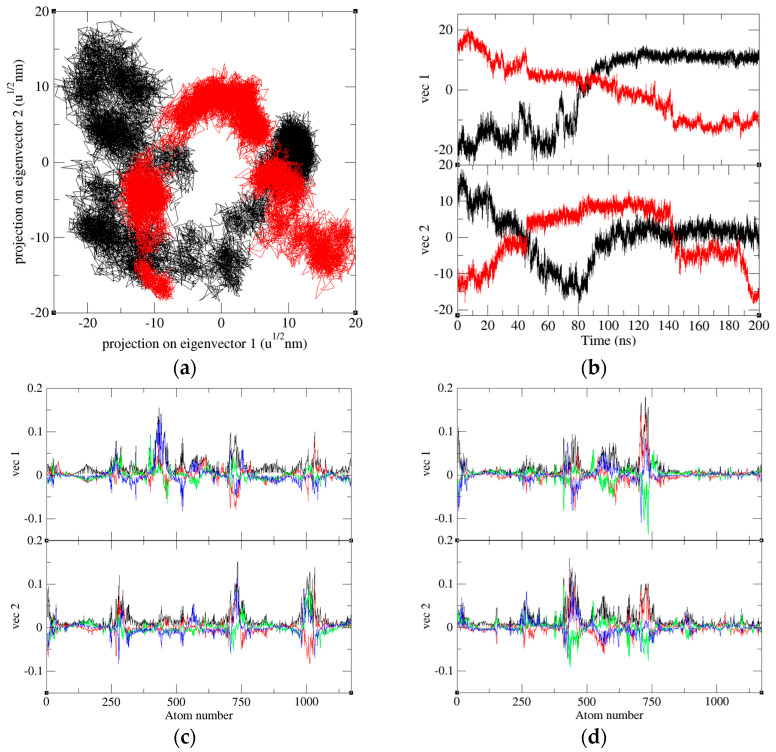
The plot of eigenvectors with eigen components. (**a**) The 2D projections of trajectories on eigenvectors of PD-1 (black) and PD-1 in the PD-1/Pembrolizumab complex (red). (**b**) The projections of trajectories on eigenvectors of PD-1 (black) and PD-1 in complex (red) at different time scale. Eigenvector components were further calculated for (**c**) PD-1 and (**d**) PD-1 in the PD-1/Pembrolizumab complex. The eigenvector components pertaining to the total, x-axis, y-axis, and z-axis were represented by the colors black, red, green, and blue, respectively.

**Figure 11 ijms-24-10684-f011:**
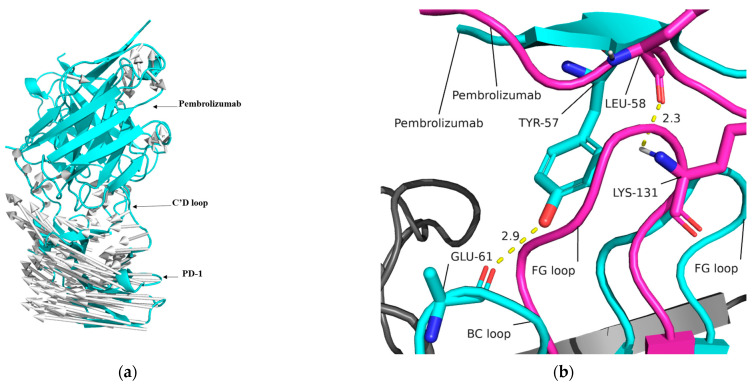
(**a**) The rotation of PD-1 during MD simulation. The arrows (silver) describe the rotation direction. (**b**) The FG loop at 200 ns (magenta) occupy the binding site of BC loop at initial state (cyan) during the rotation.

**Table 1 ijms-24-10684-t001:** The main residues participated in the hydrogen bonds, Van Der Waals and salt bridges in the crystal structure of PD-1/Pembrolizumab complex.

Complex	Hydrogen Bonds	Van Der Waals	Salt Bridges
PD-1—Light chain of Pembrolizumab	Ser_60_-Tyr_34_, Glu_61_-Tyr_57_ and Leu_128_-Tyr_53_	Glu_61_-Gly_33_ and Glu_61_-Tyr_34_	Arg_86_-Arg_96_
PD-1—Heavy chain of Pembrolizumab	Asn_66_-Arg_102_, Gln_75_-Thr_30_, Thr_76_-Tyr_101_, Asp_77_-Ser_54_, Lys_78_-Tyr_101_, Lys_78_-Thr_33_, Ser_87_-Arg_99_, Ser_87_-Asn_59_, Ser_87_-Thr_35_ and Gly_90_-Thr_58_	Gln_75_-Thr_28,_ Lys_78_-Arg_99_, Lys_78_-Arg_102_, Arg_86_-Tyr_33_, Asp_85_-Tyr_101_, Ser_87_-Ser_95_, Ser_87_-Leu_98_, Ser_87_-Leu_100_, Gly_90_-Gly_57_ and Gly_90_-Asn_59_	Asp_85_-Arg_99_

**Table 2 ijms-24-10684-t002:** Percentage of average secondary structure elements present in PD-1, and PD-1 in PD-1/Pembrolizumab complex during MD simulations.

Protein	Structure	Coil	β-Sheet	β-Bridge	Bend	Turn	α-Helix	3-Helix
PD-1	57%	25%	49%	1%	17%	7%	0%	1%
PD-1/Pembrolizumab complex	61%	25%	50%	0%	13%	11%	0%	1%

## Data Availability

Not applicable.

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
