# Peer review of "Investigation of Molecular Interactions Mechanism of Pembrolizumab and PD-1"

_ijms, 2023, doi:10.3390/ijms241310684_

Round 1
Reviewer 1 Report
In the manuscript “Investigation of Molecular Interactions Mechanism of Pembrolizumab and PD-1”, the authors have briefly described the interaction mechanism of antibody and PD-1. The insight into the interaction between PD-1 and PD-L1 may contribute to future research on PD-1 and the interaction methods of the antibody Pembrolizumab.
I will suggest a few changes.
1- The abstract should include a limitation of the finding or tools.
2- Figure 1 legend should be elaborated and a few more lines should be added.
3- The format of section 2.8 should be non-Italic.
4- It is suggested to cite a few related works on Pembrolizumab and PD-1 and related in silico studies:
doi: 10.3390/molecules24061190
doi: 10.3389/fphar.2017.00561
doi: 10.2174/1381612822666151125000550
Author Response
Response 1: A limitation of work has been included in Abstract.
Response 2: Fig 1 legend has been elaborated.
Response 3: The format of section 2.8 has been corrected.
Response 4: These papers have been cited in discussion section.
Reviewer 2 Report
The manuscript entitled “Investigation of Molecular interactions mechanism of Pembrolizumab and PD-1” investigated the effect of Pembrolizumab on the conformational changes of PD-1 using extensive molecular modeling and simulation approaches. The computational studies were well performed by the authors. The manuscript is well written and informative. I have following minor concerns.
1. Line 67: “This information is of great importance……...” Authors needs to improve this para.
2. How authors decided the time scale of MD simulation, such as 200ns performed in this study?
3. Figure 9: What does PC1 and PC2 means?
4. Be consistent with sub-figure numbers. For example, Figure 1 and figure 2 has capital letters i.e. A, B, and C, etc. while figure 3 onward have small letters i.e. a, b, and c, etc.
5. Minor typos and grammatical errors need to be checked.
6. The abbreviation should be thoroughly checked in the manuscript.
Author Response
Response 1: This paragraph has been modified.
Response 2: A general 100-200 ns MD simulations is sifficient to analyse the results. Initially we performed 100 ns MD simulation and then extended upto 200 ns.
Response 3: PC1 and PC2 has been explained in section 2.7.
Response 4: It has been modified. Thank you so much.
Response 5: The grammatical errors have been corrected throughout the manuscript.
Response 6: The abbreviation has been thouroughly checked and corrected.
Reviewer 3 Report
Wang et.al., performed structural analysis of pembrolizumab and PD-1 complex. It is a good study but for further consideration I have following concerns
Major comments
1) In result section 2.1 authors analyzed the structure of PD-1 complex using modelling but its structure is known. Authors mention modelled missing atoms. How many atoms are missing? And how is it corrected.
2) Authors used detailed computation methods, specifically molecular dynamics simulations to dissect molecular mechanism of the complex. But the message is not clear in the manuscript. Too many details are given about structure it makes difficult to understand for biologists.
3) I recommend authors to clearly convey the biological interpretation so that it will help biologist.
Author Response
Response 1: Dear Reviewer. Thank you for asking this question. When we download the PDB structure, it is common that sometimes there are several missing residues and atoms. In our case, there were 30 atoms missing from the structure. It was fixed by the modeler. Sometimes we ignore the –H and missing amino acid residue when running MD simulations using commands such as –ignh or –missing etc. It was not important to mention in the manuscript.
One more thing to mention here. We analyzed the detailed 3D structure using PDBsum as mentioned in the material methods section. PDBsum is a database that provides an overview of the contents of each 3D macromolecular structure deposited in the Protein Data Bank. The references we cited already have detailed protocols. These are basic steps for preparing inputs.
Anyway, as per your important suggestion, we have added a line in the manuscript section 2.1 as “The complex structure of PD-1 and Pembrolizumab was obtained from the Protein Data Bank (PDB: 5B8C) and missing atoms were added to the structure.”
Response 2 & 3: Dear Reviewer, thank you for asking this question. A detailed explanation of RMSD, RMSF, and RG has been added in new material and methods sections 4.2, 4.3, and 4.4. We have also explained how it is important for Protein structure analysis. Also, several explanations and interpretations of bioinformatics have been added to section 2.4.